# Sliding Window and Pseudo-labeling techniques for Cellular Segmentation

Truong N.Bui[1], Nam T.Nguyen[2], Tuyen T.Dam[3], Hanh T.Le[4], Anh K.N.Nguyen[5],
Minh H.Nguyen[6], Kien T.Le[7], Duong H.Le[8], Anh C.H.Nguyen[9], Anh N.Nguyen[10], Duong H.Nguyen[11]

**Train4Ever**
Viettel Group, Hanoi, Vietnam

{truongbn2[1], namnt54[2], tuyendt23[3], hanhlt87[4], anhnkn[5],
minhnh42[6], kienlt27[7], duonglh9[8], anhnch2[9], anhnn99[10], duongnh34[11]}@viettel.com.vn

## Abstract

Cell segmentation is a fundamental task in biomedical image analysis, which involves the identification and separation of individual cells from microscopy images. Large-size images and unannotated data are two canailing problems degrading the performance in cell segmentation. Regarding these issues, we propose sliding window and pseudo-labeling techniques by conducting several experiments on different neural architectures. Following this approach, our method achieves a significant performance improvement and a final result of 0.8097 F1 score on the tuning set and 0.6379 F1 score on the test set of Weakly Supervised Cell Segmentation in Multi-modality Microscopy challenge hosted at NeurIPS 2022.

## 1 Introduction

Cell segmentation is crucial for biomedical research. Cells provide structure and function for all living things and are considered the smallest form of life. A lot of diseases or disorders such as meningitis, malaria, diabetes, a type of cancer, cystic fibrosis, or Alzheimer's disease are caused by problems at a cell or molecular level. Physical damage such as a burn or broken bone also causes damage at the cell level. By understanding the cell activities and mechanisms, cell biologists can determine the issue and are able to find effective treatment. The first step in understanding cells is to detect and localize the contour of each cell instance. But manual cell segmentation is costly and labor-consuming. Accurate instance segmentation of these cells with the help of computer vision could lead to new and effective discoveries to treat the millions of people with there above disorders.

In this competition, cell segmentation consists of two main problems: the limited amount of annotated images and image size variation. Hence, our approach has two main techniques to directly tackle these problems and can be summarized as follow:

- We apply the sliding-window method to divide large images into smaller patches which helps maintain the original resolution and enhance the model's performance.

- We implemented pseudo-labeling on unlabelled images for data enrichment.

- We also conducted experiments measuring the performances of different approaches including anchor-free and anchor-based models on this cell segmentation dataset.

36th Conference on Neural Information Processing Systems (NeurIPS 2022).

## 2 Method

### 2.1 Sliding Window

One of the main problems we need to handle in this competition is the variety of image sizes. Considering the tuning set, the image sizes vary from the largest 10496 x 8415 to the smallest 591 x 447. Downsizing the large images to a fixed smaller size decreases the image resolution, causing the model performance to fall remarkably. Thus, we apply the sliding-window method, i.e., dividing a large image into smaller patches.

The sliding window technique helps maintain the original resolution of the images. However, it still has two drawbacks:

- The inference time is longer and increases quadratically with the length of the image edge. Nevertheless, since the tolerance time also scales quadratically with the length of the image edge, the inference time remains lower than the tolerance time. Consequently, it does not affect the running time rank.
- Our sliding window is non-overlapped, which means every pixel appears in only one patch. This is the simplest and costs the least computation implementation but it suffers what we call the edge effect: the phenomenon that cells lying on different patches are cut into smaller fragments and ultimately recognized as many different cells which leads to the model precision decrease.

Currently, our patch size selection algorithm is completely heuristic and described by the algorithm 2.1. Algorithm 2.1 parameters including threshold values and corresponding patch sizes were determined by tuning on our validation dataset. Hence, this algorithm can be highly biased and prone to this dataset. Finally, the overall sliding window technique is represented in the algorithm 2.1. In the 2.1, two parameters $t_{size}$ and $t_{cell}$ denote the minimum thresholds of the shortest edge and the number of instances predicted initially. These values were 4000 pixels and 5 instances, selected by tuning on the validation dataset

---

**Algorithm 1** Patch size selection

---

    **Input**: $w, h$ - the width and height of the image, respectively.
    **Output**: the patch size

 1: $\alpha = min(w, h)$: the shortest edge
 2: **if** $\alpha < 2000$ **then**
 3:     **return** 1024
 4: **else if** $\alpha \geq 2000$ and $\alpha \leq 3000$ **then**
 5:     **return** 256
 6: **else if** $\alpha \geq 3000$ and $\alpha < 4000$ **then**
 7:     **return** 512
 8: **else if** $\alpha \geq 4000$ and $\alpha < 15000$ **then**
 9:     **return** 1024
10: **else**
11:     **return** 2048
12: **end if**

---

**Algorithm 2** Sliding window

---

    **Input**: $w, h$ - the width and height of the image, respectively; $t_{size}$ - the threshold image sizes; $t_{ncell}$ - the threshold number of cells.

 1: $n_{cell}$: the number of cells of the prediction without applying the sliding window technique.
 2: $\alpha = min(w, h)$: the shortest edge
 3: **if** $\alpha \geq t_{size}$ or $n_{cell} < t_{ncell}$ **then**
 4:     Applying Algorithm 2.1 in order to find the patch size.
 5:     Dividing the image into non-overlapped patches.
 6:     Feeding these patches into the network and merging all predicted instances.
 7: **end if**

---

## 2.2 Pseudo Labeling

The training set contains only 1000 labeled images, but more than 1500 images are unlabelled. Hence, we can leverage the original dataset by implementing Pseudo-labelling. The main idea of this approach is to use the predictions of a trained model as the ground truth annotations, which are ultimately appended to the original dataset to push more information into the data set. Furthermore, the Pseudo-labelling method can also be used as an ensemble method compressing multiple models' knowledge into a single model. The pipeline of pseudo-labeling will be described in detail in section 3.4.

## 3 Experiments

### 3.1 Datasets and pre-trained models

The provided dataset contains 1000 labeled images with four microscopy modalities. The distribution of each modality is illustrated in table 1. The unlabeled dataset which contains more than 1500 images is used for the pseudo-labeling task. We also use the external LIVECell dataset [1].

Our pre-trained models are Cascade Mask RCNN Resnest200 and Cellpose, pre-trained on the LIVECell dataset, CBNetv2 and CenterMask pre-trained on the COCO dataset [2].

Table 1: Number of images per modality

| Modality | Number of images |
|---|---|
| Brightfield | 300 |
| Fluorescent | 300 |
| Phase-contrast | 200 |
| Differential interference contrast | 200 |

### 3.2 Environment settings

The development environments and requirements are presented in Table 2.

Table 2: Development environments and requirements.

| Environment | Specification |
|---|---|
| CPU | Intel(R) Core(TM) i9-7900X CPU@3.30GHz |
| RAM | 128GB |
| GPU (number and type) | One NVIDIA V100 32G |
| CUDA version | 11.1 |
| Programming language | Python 3.7.11 |
| Deep learning framework | Pytorch (Torch 1.10.1, torchvision 0.11.2) |

### 3.3 Model Experiments

We conducted experiments on different models with different architectures: anchor-based: Cascade Mask RCNN Resnest200 [3], CBNetv2 [4], anchor-free: CenterMask [5] and cell-specific algorithm CellPose [6]. The results are illustrated in Table 3. We observed that anchor-free models like CenterMask struggled to detect small or non-convex cells and have much lower recall than anchor-based models; CellPose outperformed other models in images with fluorescent modality but was not stable in other modalities and required many hyperparameter-tuning and we were short in time and cannot spend more effort for cellpose. Consequently, we only focused on improving Resnest200 and CBNetv2.

We observed that anchor-free models like CenterMask struggled to detect small or non-convex cells and have much lower recall than anchor-based models; CellPose outperformed other models in images with fluorescent modality but was not stable in other modalities and required many hyperparameter-tuning. Although these experiments were conducted without the sliding window technique and

the results might only indicate how well these models performed with our default settings, the differences between CenterMask and the other two models Resnest200 and CBNetv2 are significant (0.1002 and 0.1304 respectively) and overwhelm the bias of our settings toward models performance. Consequently, we only focused on improving Resnest200 and CBNetv2. Our detailed implementation will be described in sections 3.3.1, 3.3.2, 3.3.3, 3.3.4.

Table 3: F1 scores on Tuning Set

| Model | F1 (w/o sliding window) |
|---|---|
| CenterMask | 0.6405 |
| Cascade Mask RCNN Resnest200 | 0.7407 |
| CBNetv2 | 0.7709 |
| CellPose | 0.7805 |

### 3.3.1 CBNetV2

In our approach, we leveraged the CBNetV2 repository by instantiating the CBNetV2 model with a Swin-transformer backbone. We selected $K = 2$ as the hyperparameter value, in order to balance the need for fast inference time and high accuracy. Notably, we utilized image input sizes from (780,1333) and (1100, 1333) during training. Our hyperparameter settings are illustrated in table 4
.

Table 4: CBNevtV2 hyperparameters

| hyperparameter | value |
|---|---|
| number of nms input regions | 4000 |
| max number of nms output regions | 3000 |
| nms: iou-threshold | 0.8 |
| mask binary score threshold | 0.3 |
| nms: iou-threshold | 0.3 |
| max regions per image | 1000 |

### 3.3.2 Resnest200

We use Detectron2-ResNeSt to implement Cascade Mask RCNN Resnest200 model. To improve the accuracy of our model on current dataset, we conducted experiments with different pretraining options (pretrained on imagenet, train from scratch, and pretrained Livecell) and found that using a ResNeSt200 model pre-trained on the LIVECell dataset, yielded superior performance. During the training phase, we set the minimum image size to 440 pixels. Detailed hyperparameters are illustrated in table 5.

Table 5: Resnest200 hyperparameters

| hyperparameter | value |
|---|---|
| PRE_NMS_TOPK_TRAIN | 12000 |
| POST_NMS_TOPK_TRAIN | 3000 |
| PRECOMPUTED_PROPOSAL_TOPK_TEST | 1000 |

### 3.3.3 CenterMask

We choose the version Centermask-lite V39 to balance the inference time and the accuracy. Hyperparameters are illustrated in Table 6.

The model achieves a good score on the valid dataset however it does not generalize well on the tuning set. It performed poorly on the non-convex cell and its precision is lower than the anchor-based model due to the variety of cell shapes. Additionally, it faces a severe problem of overlapping cells in cases where cells are close to each other. In each modality, it faces the same problem as the anchor-base model, especially on the modality DIC due to the noisiness in the dataset and in the fluorescent where the cell's appearances are dense. The visualization was shown in Figure 1 and 2.

Table 6: Centermask hyperparameters

| hyperparameter | value |
|---|---|
| detections-per-image | 1600 |
| post-nms-topk-test | 5000 |
| post-nms-top-train | 2000 |

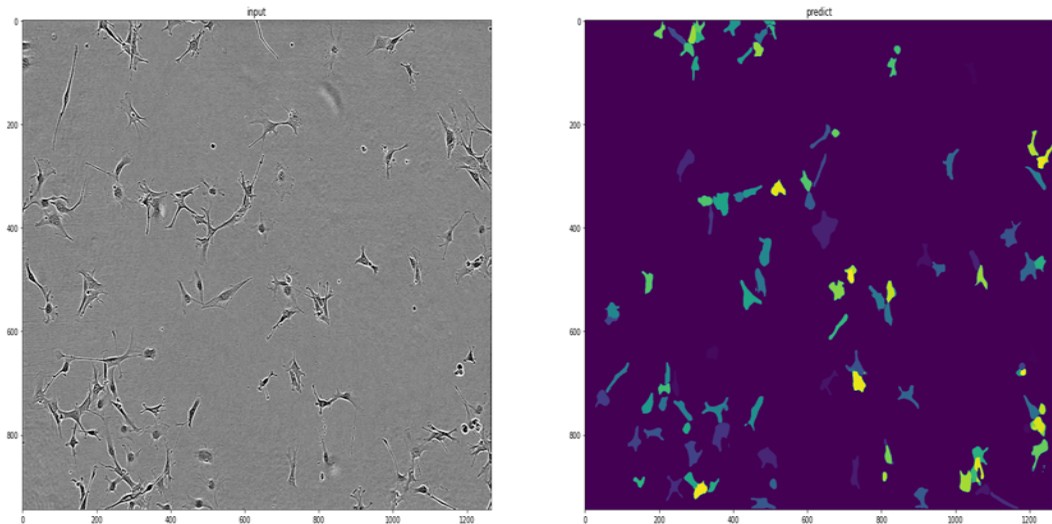

Figure 1: Centermask recognized a non-convex cell as many smaller and more convex cells

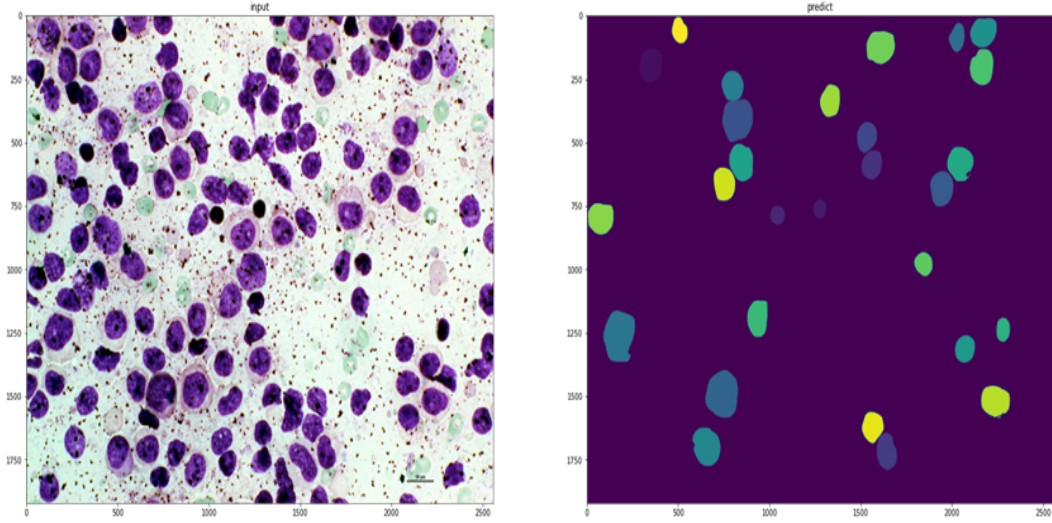

Figure 2: Centermask has low recall on "easy images", which were well recognized by other models.

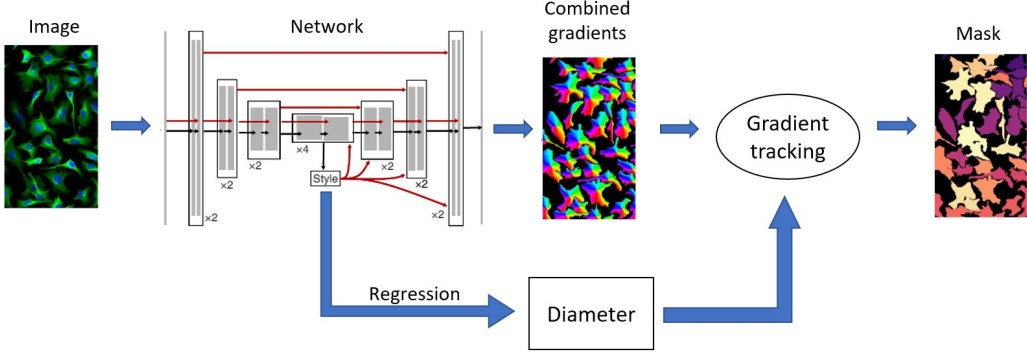

Figure 3: Cellpose inference pipeline

### 3.3.4 CellPose

Cell pose is generalist models for cellular segmentation pre-trained on large datasets such as TissueNet, and LiveCell. Cell pose is a segmentation base model that uses Unet as a backbone to extract features from 1 channel input image. The neural network was trained to predict three outputs: horizontal and vertical gradient of the topological maps, a binary map that indicate if a given pixel is inside or outside a cell. Topological maps were generated from the instance segmentation label. The Center of each cell is computed as the median of all pixels inside the cell. After that, simulated diffusion was calculated for all pixels inside the cell, the value based on the distance to the cell center. Vertical and horizontal gradients are finally calculated on simulated diffusion as the two outputs of the model. Cell pose using the output feature in the immediate layer and call it style to predict the mean diameter of the cell in the picture. Cell pose uses a binary segmentation map to filter all inside cell pixels and then combined gradients for each inside cell pixel is an aggregate vector from the horizontal and vertical gradient. Gradient tracking is an algorithm that finds a center cell for each pixel, all pixels belonging to a given cell can be routed to its center. In the end, all pixels of each cell are grouped, identified by this ID, and made into a final output instance segmentation mask. The overview of training and inference flow is described in figure 3:

We trained the cellpose segmentation model using the Livecell Blue channel for 100 epochs, and used that as a pre-trained size model and then trained size model for 15 epochs.

### 3.4 Pseudo-labeling Pipeline

### 3.4.1 Downsample unlabeled dataset

According to our experience, the number of pseudo-labeled images should not exceed the number of labeled images. Besides, we observed duplicated patterns in many images, therefore a much smaller number of images should be well representative of the whole dataset. Following that consideration, we clustered the dataset by using bit-wised distance in hashed-image vector space and randomly picked out a small fragment of images from each cluster. Ultimately, we received an additional 487 unlabeled images.

### 3.4.2 Allocate dataset to models

We trained our most three competitive models which are CBNetv2, Cascade Mask RCNN Resnest200, and Cellpose on the labeled dataset (not including the external datasets). In the validation dataset, we created some metadata which are combinations of image size and modality criteria, then divided images into smaller partitions based on that metadata. By doing so, we figured out which model gave the best validation results with particular metadata and used that model to make predictions on the unlabeled dataset.

We then proceeded to filter out instances (cells) within an image that had low confidence scores and kept only instances with a confidence score greater than a threshold, which we choose to be 0.9. We

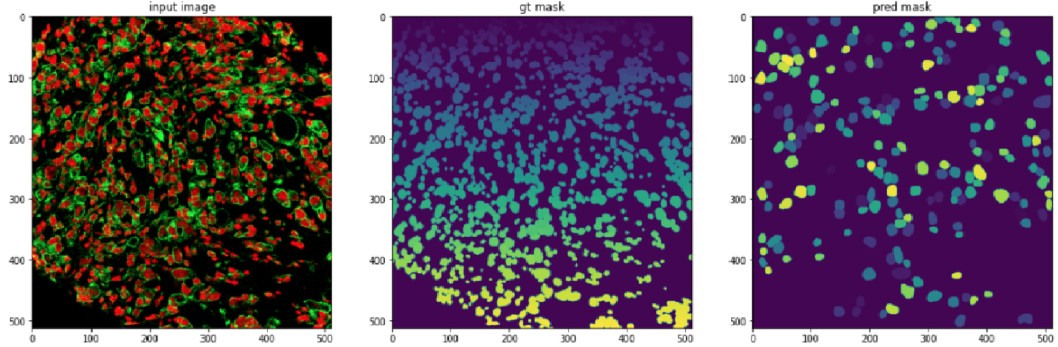

Figure 4: Low recall image, 0.24 F1 score

suspect that deciding the value of this threshold might be crucial; a low value can create many false positive instances and on the opposite, a high value can create false negative instances, both will mislead the model. Finally, we inserted the processed pseudo-labeled images into the original dataset to retrain our model.

## 4 Results and discussion

### 4.1 Quantitative results on tuning set

Table 7: Results on Tuning Set (F1)

|  | CBNetv2 | Cascade Mask RCNN Resnest200 |
|---|---|---|
| w/o Sliding Window, Pseudo-labeling | 0.7709 | 0.7407 |
| w/o Sliding window | 0.7205 | 0.7158 |
| w/o Pseudo-labeling | 0.8097 | 0.7971 |
| Sliding window, Pseudo-labeling | 0.7635 | 0.7497 |

Sliding window consistently enhanced the model's performance by 0.0388 and 0.0564 F1 scores with CBNetv2 and Cascade Mask RCNN Resnest200 respectively.

Whilst, our attempt to utilize unlabeled data was unsuccessful, inserting the unlabeled images made the F1 score drop significantly. We only trained on the pseudo-labeled dataset once and did not have the chance to correct any mistake that might occur, such as picking the threshold to filter instances we mentioned in section 3.4.2.

Surprisingly, the more cumbersome and pre-trained on the microscopic dataset Cascade Mask RCNN Resnest200 was consistently outperformed by CBNetv2, which was only pre-trained on COCO dataset.

### 4.2 Qualitative results on validation set

One of the most challenging problems of cell segmentation task is high cell density images where cells are not separated. Figure 4 illustrates the model struggled to segment cells in high cell density images, which has fluorescent modality. In figure 5, having the same fluorescent modality but the cell density is lower, our model achieved a much higher F1 score.

Figure 6 illustrates the effect of image resizing. Implementing sliding windows increases the resolution of the image in comparison with not using sliding windows. In this case, the model recognized the same image as two completely different types of cell. Thus, we believe that an ideal sliding window strategy should be adaptive and dynamically change relative to cell type.

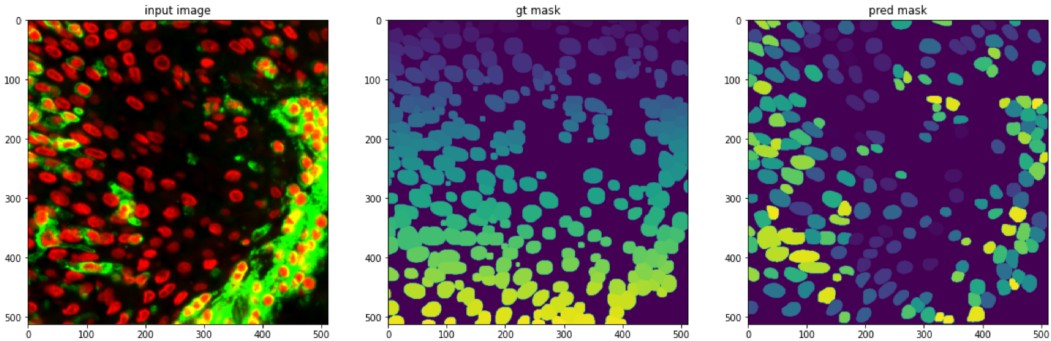

Figure 5: High recall image, 0.82 F1 score

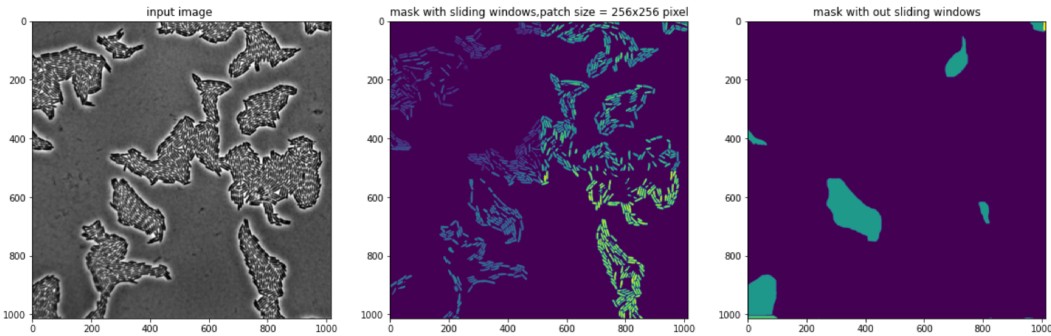

Figure 6: Effect of image resizing

## 4.3 Segmentation efficiency results on validation set

Having a limited memory resource (10 GB GPU RAM while inferring), we had to prevent any redundant usage. Initially, the segmentation mask is presented as a 3D tensor with the shape $C \times W \times H$ which is the default implementation of mmdetection library [7] where $C$ is the number of instances, and $W$ and $H$ are the width and the height of the image respectively, we paste instance by instance into a $W \times H$ mask. This can lead to cell overlapping but cost $C$ times less memory.

The compact CBNetv2's runtime is much faster than Cascade Mask RCNN and able to infer all the images in the test dataset within the tolerance time while having better performance. Considering the effect of different types of images, we observed no patterns about which images cost more runtime than the others. We suspect the differences in runtime were completely random.

Table 8: Number of tolerance time exceeded images on Tuning Set

| Model | Number of images |
|---|---|
| CBNetv2 | 4 |
| Cascade Mask RCNN Resnest200 | 12 |

## 4.4 Testing set result

There are significant differences between performances in the tuning set and test set, the F1 scores dropped from 0.8097 to 0.6379. The Fluorescent modality was far below and dragged down the overall average. We suspect that the differences were caused by the fact that our model was insufficiently adapted to different sizes of the images especially the fluorescent images which have cells that vary in size and shape.

Table 9: Testing result

| Modality | Mean F1 score |
|---|---|
| All | 0.6379 |
| Brightfield | 0.8495 |
| Fluorescent | 0.249 |
| Phase-contrast | 0.7732 |
| Differential interference contrast | 0.7217 |

### 4.5 Limitation and future work

The model performance is extremely affected by the cell size and consequently affected by the image size. Currently, we implemented a rule-based algorithm to choose patch size for sliding window algorithm based on the image size. But we observed the patch size should be chosen not by the image size but the cell size. Hence, we believe a sub-model which predicts the mean cell size within an image to decide the patch size should enhance the performance.

Another limitation of our approach is the edge effect that we mentioned in section 2.1. The edge effect can be resolved by implementing an overlapped sliding window, which is relevant to the stride convolutional layer. Instead of dividing images into non-overlapped patches, now a pixel can be captured in different patches and consequently, it increases the probability of the whole cell appearing in only one patch and can be segmented as no edge effect exists. But on the other hand, it will also cost redundant computation of feeding the same part of images multiple times. In addition, this approach will require an algorithm to merge the mask of the patches, one possible solution is the Non-maximum Suppression algorithm. Although this approach can resolve the edge effect, since only a minor number of cells lie on the edges, it may not be worth the additional computation. An alternative solution to the edge effect that does not require unnecessary computation as the overlapped sliding window is to identify different fragments that belong to the same cell but this task is far from trivial.

## 5 Conclusion

We conducted experiments on image cell segmentation with two techniques: sliding windows and pseudo-labeling. The sliding windows method gained a significant performance improve, but on the other hand this method is unstable and depends on hyperparameter tuning. The pseudo-labeling method did not bring in any performance improvement and need further tuning.

## 6 Acknowledgement

Our implementation for participation in the NeurIPS 2022 Cell Segmentation challenge has not used any private datasets other than those provided by the organizers and the official external datasets and pre-trained models. The proposed solution is fully automatic without any manual intervention.

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
