# OpenReview forum: "Sliding Window and Pseudo-labeling techniques for Cellular Segmentation"
_NeurIPS.cc/2022/Challenge/CellSeg — Submitted to NeurIPS CellSeg 2022_

### Official Review · Reviewer_oojU · 2022-12-25
**Unclear method**

**Rating:** 4
**Confidence:** 4

**Review:**

The paper investigates the use of several models for cell segmentation, but the presentation of the methods and description of other approaches is unclear. Overall, the authors investigate the use of 4 different models for cell segmentation. After initial application, they pick 2 models for which the performance is further investigated when using a sliding window approach for inference and pseudo-labeling.

Some aspects of the approaches are not described in enough detail:
- The sliding window approach is only described in minimal detail. Do you use overlapping windows or no overlap? The patch size selection from Listing 1 seems implausible: for "shortest_edge < 2000" a patch size of 512 is chosen (second if), whereas for "2000 <= shortest_edge < 3000" a (smaller!) patch size of 256 is chosen (first if). What is the rationale?
- The description of pseudo-labeling is lacking details: how exactly are the pseudo-labels used to train a new model? Do you train a new model on labeled data and unlabeled data with pseudo-labels or fine-tune on the pseudo labels? Which representation of the pseudo-labels is used?
- Can you describe the procedure in "3.4.2 Allocate datasets to models" in more details? Do you train a separate model for each "partition" mentioned in this paragraph? This approach is not mentioned anywhere else in the paper, but seems quite crucial. Please describe in more detail and elaborate on the consequences of model performance.

A further methodological question:
- To my understanding the models in Table 2 are applied without the sliding window approach, Are the input images resized to fit the models? As the authors also remark, this can lead to inferior performance. It may further rely on how the models deal with large images internally, so I am not convinced that this comparison is fair. I would recommend to compare all models with the sliding window approach as well.

---

> ### Author Response · Authors · 2023-02-21
> **Response to Reviewer oojU**
>
> **Question 1**: The sliding window approach is only described in minimal detail. Do you use overlapping windows or no overlap? The patch size selection from Listing 1 seems implausible: for "shortest_edge < 2000" a patch size of 512 is chosen (second if), whereas for "2000 <= shortest_edge < 3000" a (smaller!) patch size of 256 is chosen (first if). What is the rationale? \
> **Answer**:
> - The details of the sliding window method are described by Algorithm 2. We use non-overlapped windows.
> - The patch size selection: Threshold values and corresponding patch sizes were just determined by tuning on our validation dataset.
>
>
> **Question 2**: The description of pseudo-labeling is lacking details: how exactly are the pseudo-labels used to train a new model? Do you train a new model on labeled data and unlabeled data with pseudo-labels or fine-tune on the pseudo labels? Which representation of the pseudo-labels is used?\
> **Answer**:
> - According to our experience, the number of pseudo-labeled images should not exceed the number of labeled images. Besides, we observed duplicated patterns in many images, therefore a much smaller number of images should be well representative of the whole dataset. Following that consideration, we clustered the dataset by using bit-wised distance in hashed-image vector space and randomly picked out a small fragment of images from each cluster. Ultimately, we received an additional 487 unlabeled images.
> - We divided the obtained images into smaller partitions by creating modality metadata. With each modality type, we use the model that gave the best validation results on that modality to predict instances. We then proceed to filter out instances with low confidence scores and keep only instances with confident score > threshold, with threshold = 0.9. Finally, we fine-tuned our model on a merged dataset consisting of labeled data and unlabeled data with pseudo-labels.
>
>
> **Question 3**: Can you describe the procedure in "3.4.2 Allocate datasets to models" in more details? Do you train a separate model for each "partition" mentioned in this paragraph? This approach is not mentioned anywhere else in the paper, but seems quite crucial. Please describe in more detail and elaborate on the consequences of model performance.\
> **Answer**: We trained every models which are CBNetv2, Cascade Mask RCNN Resnest200, and Cellpose on the whole labeled dataset. Partition were created from the validation dataset based on combinations of image size and modality criteria.  By doing so, we figured out which model gave the best validation results with particular metadata and used that model to make predictions on the unlabeled dataset. As we mentioned in the 4.1. Quantitative results on tuning set, training on the pseudo-labeled data did not bring any improve but decreased on models performance. But we have only done the pseudo-labeling once and did not have the chance to correct any mistake that might occur. Hence, it is hard for us to make further analysis about our pseudo-labeling pipeline.
>
> Question 4: To my understanding the models in Table 2 are applied without the sliding window approach, Are the input images resized to fit the models? As the authors also remark, this can lead to inferior performance. It may further rely on how the models deal with large images internally, so I am not convinced that this comparison is fair. I would recommend to compare all models with the sliding window approach as well. \
> Answer: Yes, models in Table 2 were applied without the sliding window.  Although these experiments were conducted without the sliding window technique and the results might only indicate how well these models performed with our default settings, the
> differences between CenterMask and the other two models Resnest200 and CBNetv2 are significant (0.1002 and 0.1304 respectively) and overwhelm the bias of our settings toward models performance and we believe that it is reasonable to save time for the competition and ignore the risk of leaving out good models.

---

### Official Review · Reviewer_wjco · 2022-12-28
**Incomplete description**

**Rating:** 4
**Confidence:** 4

**Review:**

This paper reports on the effectiveness of window-scoring methods and pseudo-labeling techniques for semi-supervised cell segmentation task. The model used achieves good performance on the validation set, addressing the challenges of excessive image resolution and how to use unlabeled images. However, I consider that more details should be added to this paper：

1. How does the sliding window method affect the final result? How are the hyperparameters determined?

2. How is the pseudo-labeling method implemented? How is the model trained to be updated?

3. May the introduction and the results of the test be more detailed?

---

> ### Author Response · Authors · 2023-02-21
> **Response to Reviewer wjco**
>
> **Quesion 1**: How does the sliding window method affect the final result? How are the hyperparameters determined?\
> **Answer 1**:
> - Sliding window consistently enhanced the model’s performance by 0.0388 and 0.0564 F1 scores with CBNetv2 and Cascade Mask RCNN Resnest200, respectively.
> - The hyperparameters were just determined by tuning on our validation dataset.
>
>
> **Question 2**: How is the pseudo-labeling method implemented? How is the model trained to be updated?\
> **Answer 2**: We divided the obtained unlabeled images into smaller partitions by creating modality metadata. With each modality type, we use the model that gave the best validation results on that modality to predict instances. We then proceed to filter out instances with low confidence scores and keep only instances with confident score > threshold, with threshold = 0.9. Finally, we fine-tuned our model on a merged dataset consisting of labeled data and unlabeled data with pseudo-labels.
>
> **Question 3**: May the introduction and the results of the test be more detailed?\
> **Answer 3**: Test result details are updated in section 4, the new version.

---

### Official Review · Reviewer_HgTd · 2022-12-31
**Well writen but it is somewhat digress.**

**Rating:** 5
**Confidence:** 4

**Review:**

Summary:
The paper reports the performances of some cell segmentation methods and the effectiveness of sliding windows and pseudo labels on the CBNetv2 and Cascade Mask RCNN Resnest200.

Strength:
- Details of multiple methods are provided, with clear descriptions.

Comment:

They report the results of multiple methods but observe that the sliding window is the one that improves their results but the pseudo labels. However, there is a lack of information on this key method that improves their outcomes.
- There is only one rule (get_patch_size function) to conducting the sliding window is provided. What happens with the images with the shortest_edge < 2000? Are they being used the way they are?
- How are the hyperparameters on this "def get_patch_size ( shortest_edge )" determined?
- Implementation details of CBNetv2 and Cascade Mask RCNN Resnest200 needed? What is the input size?

---

> ### Author Response · Authors · 2023-02-21
> **Response to Reviewer HgTd**
>
> **Question 1**: There is only one rule (get_patch_size function) to conducting the sliding window is provided. What happens with the images with the shortest_edge < 2000? Are they being used the way they are?\
> **Answer 1**:
> - The details of the sliding window method is described by Algorithm 2, we use two rules to decide to apply the sliding window.
> - The images with the shortest_edge ( the minimum between width and height) < 2000 are applied sliding window with patch_size =1024 (algorithm 1) when a number of predicted instances < 5, otherwise not apply.
>
> \
> **Question 2**: How are the hyperparameters on this "def get_patch_size ( shortest_edge )" determined?\
> **Answer 2**:
> We make this function clearer in algorithm 1, get_patch_size algorithm is completely heuristic since threshold values and corresponding patch sizes were just determined by tuning on our validation dataset.
>
> \
> **Question 3**: Implementation details of CBNetv2 and Cascade Mask RCNN Resnest200 needed? What is the input size?\
> **Answer 3**:
> - In our approach, we leveraged the CBNetV2 repository, accessible via the following link: https://github.com/VDIGPKU/CBNetV2, to instantiate the CBNetV2 model with a Swin-transformer backbone. Specifically, we selected K = 2 as the hyperparameter value, in order to balance the need for fast inference time and high accuracy. Notably, we utilized image input sizes from (780,1333) and (1100, 1333) during training.
> - We use Detectron2-ResNeSt for implement cascade Mask RCNN Resnest200 model, which can be accessed via the following link: [https://github.com/chongruo/detectron2-ResNeSt] To improve the accuracy of our model on current dataset, we conducted experiments with different pretraining options (pretrain on imagenet, train from scratch, and pretrain Livecell) and found that using a ResNeSt200 model pretrained on the LIVECell dataset, which is publicly available at [https://github.com/sartorius-research/LIVECell], yielded superior performance. During the training phase, we set the minimum image size to 440 pixels.
> - For more detailed hyper parameters for both models, you can find it in our paper after the revision.

---

### Decision · Program_Chairs · 2023-01-19

Accept